# Functional neuronal circuits emerge in the absence of developmental activity

**Dániel L. Barabási** [1,2,3] ✉, **Gregor F. P. Schuhknecht** [1,3] **& Florian Engert** [1]

The complex neuronal circuitry of the brain develops from limited information contained in the genome. After the genetic code instructs the birth of neurons, the emergence of brain regions, and the formation of axon tracts, it is believed that temporally structured spiking activity shapes circuits for behavior. Here, we challenge the learning-dominated assumption that spiking activity is required for circuit formation by quantifying its contribution to the development of visually-guided swimming in the larval zebrafish. We found that visual experience had no effect on the emergence of the optomotor response (OMR) in dark-reared zebrafish. We then raised animals while pharmacologically silencing action potentials with the sodium channel blocker tricaine. After washout of the anesthetic, fish could swim and performed with 75–90% accuracy in the OMR paradigm. Brain-wide imaging confirmed that neuronal circuits came 'online' fully tuned, without requiring activity-dependent plasticity. Thus, complex sensory-guided behaviors can emerge through activity-independent developmental mechanisms.

Understanding how functional neuronal circuits are established during development is a fundamental challenge in neuroscience. One prominent line of thinking is that network structure is initially only rudimentarily defined by genetic mechanisms, and that functional wiring is established through activity-dependent processes[1-3]. Indeed, research in artificial intelligence and computational neuroscience have highlighted how weakly structured models can learn to perform complex tasks, even outperforming humans at Chess and Go[4]. Such models are thought to abstract the biological phenomenon of experience-dependent rewiring of neuronal circuits, which is assumed to be critical for the maturation of the functional brain[5-7].

The significant role of sensory-driven neuronal activity in brain development was made apparent by the discovery that competitive interaction between the two eyes during development is responsible for ocular column formation in cats and primates[8-10]. These seminal experiments, and many that followed, deprived animals of sensory inputs early in development and then showed major functional abnormalities in the brain[11,12]. In turn, theoretical and computational neuroscience work began to internalize the idea that stimulus-driven neuronal activity is sufficient, and at times necessary, for tuning the brain[13]. However, it is still unclear whether 'normal' sensory experience

during development is indeed necessary for the emergence of complex behaviors, or to what extent the underlying circuit modules are 'hard-wired', i.e., genetically encoded[14-16].

More recently, the discovery of correlated spontaneous neuronal activity in early brain development has illuminated the potential for activity to tune neuronal circuits prior to the arrival of sensory input[17,18]. Indeed, wave-like activity patterns in the retina have been shown to contribute to the refinement of visual circuitry, and can even encode relevant statistical properties of natural stimuli[18-21]. Yet, perturbations of such developmental activity, including sensory deprivation studies, are usually restricted to small brain regions and target only a single sensory modality, while neighboring input streams remain intact[9,22-25]. This generates competition between the perturbed modality and its unperturbed neighbors, making it impossible to distinguish whether a loss-of-function was caused by the lack of neuronal activity itself, or by a competitive takeover from other modalities.

In summary, testing whether any or all, of these activity-dependent components of development are truly necessary for the maturation of functional neuronal circuits requires the ability to reversibly block all spiking activity throughout the period of brain formation - an intervention previously lethal at birth[26,27]. Here, we have

[1]Department of Molecular and Cellular Biology, Harvard University, Cambridge, MA, USA. [2]Biophysics Program, Harvard University, Cambridge, MA, USA. [3]These authors contributed equally: Dániel L. Barabási, Gregor F. P. Schuhknecht. ✉e-mail: danielbarabasi@gmail.com

overcome this challenge by utilizing the reversible sodium channel blocker tricaine to pharmacologically block all action potentials during the four days in which the central nervous system of the larval zebrafish is formed. We found that, after tricaine washout, zebrafish could not only perform complex visuomotor behaviors, but they also exhibited fully functional and appropriately tuned neuronal cell types whose response properties were no different from those found in normally reared animals.

## Results

The early maturation of functional circuits in larval zebrafish provides a unique opportunity to study the contributions of activity-dependent and -independent components of neuronal development for vertebrate behavior. For instance, when exposed to whole-field motion, six-day-old zebrafish can turn and swim to match the direction and speed of the visual stimulus, which allows the animal to stay stationary in a moving stream[28–30]. This complex sensorimotor behavior, known as the optomotor response (OMR), is governed by multiple neuronal populations with distinct response properties and time constants that are activated across various brain regions[31,32]. Notably, their interplay encodes complex computations, ranging from motion discrimination[33,34], to evidence integration[35–38], swim-decision making[39,40], and navigation[41], which are considered hallmarks of higher cognitive function in mammals and primates. Thus, the OMR presents a rich paradigm for studying the contributions of sensory experience, neuronal activity, and activity-independent mechanisms in the maturation of a complex neuronal circuit.

In order to quantify the larval zebrafish's OMR, we utilized a custom-designed behavioral rig[37] (Fig. 1a), in which animals can be exposed to gratings that move orthogonally to their body axis (in either leftward or rightward direction) under closed-loop conditions[33,34]. When shown no stimulus, fish mainly swam forward (i.e., zero-degree turns), but also made small-angle turns to the left or right, as seen in the shoulders around the dashed line at zero degrees (Fig. 1b, blue). However, when presented orthogonally moving gratings, animals displayed a strong tendency to enhance the probability of turning in the stimulus direction, which can be observed as a "bump" to the left of zero (dashed vertical line) for leftward moving stimuli, and to the right of zero for rightward moving stimuli (Fig. 1c, blue). This asymmetry in turn statistics disappeared for forward-moving stimuli. We can summarize these patterns of response to directed whole-field motion by plotting the average cumulative turn angle for all fish under the different stimulus conditions (Fig. 1d, blue), which highlights the persistent turn preferences over the course of an experiment. In addition to turn probabilities, we can also quantify the proportion of "correct" turns over all swim bouts, which provides a sensitive summary metric for behavioral performance that is used throughout the study (Fig. 1e, blue). Lastly, we examined the overall swim activity of zebrafish larvae under all stimulus conditions, and found that their bout rate increased from 0.5 Hz in the absence of stimulation (Fig. 1b inset, blue) to 1 Hz when whole-field motion in any direction was presented (Fig. 1c inset, blue).

In order to quantify the contribution of early sensory experience to OMR, we raised zebrafish in complete darkness, thereby eliminating visual experience altogether[42]. Generally, larval zebrafish start swimming immediately after they hatch at 2-3 days post fertilization (dpf), allowing them several days of visual and motor experience prior to our quantification of OMR at 6 dpf. Thus, our dark-reared animals swam and behaved for several days in the absence of any visual input, and were first exposed to light at the moment of OMR testing. When we applied our behavioral quantification protocol to dark-reared larvae, we found no detectable difference in turn probability (Fig. 1b, c, "dark," black vs "control," blue), bout rate (Fig. 1b, c insets), or cumulative turn angle (Fig. 1d). Nevertheless, we observed a statistically significant decrease in the fraction of correct turns in dark-reared animals

(Fig. 1e); however, the small effect size (Control: 96.7%, Dark: 94.9% median correct) suggests that this is unlikely to be of ethological significance, but rather reflects the large number of fish ran for the study. Further, we considered the possibility that visual stimulation might be sufficient to rapidly entrain the circuit, in which case we would expect dark-reared fish to perform poorly at first, but improve over subsequent trials. However, we found that already at the very first trial, dark-reared animals performed with a median of 94% turn accuracy, and only slightly improved their performance over the next four trials (Supplemental Fig. 1), which is in fact consistent with the rapid behavioral adjustment that occurs when animals receive unexpected sensory feedback signals[32,43]. Either way, it is unlikely that a significant developmental restructuring of neural circuits[44] is at play when larval zebrafish are raised in darkness[45], even if subtle behavioral and neurophysiological effects can be observed when the system is specifically challenged[46,47].

We additionally considered whether irregular visual experience during development might have a deleterious effect on behavior, for which we raised fish under a constant 1 Hz strobe light during daytime[48–50]. Yet, similar to the dark-reared animals, strobe-reared fish showed no change in any of our behavioral metrics (yellow, Fig. 1b–d), except for a slightly smaller fraction of correct turns (Fig. 1e). We again found that this minor effect disappeared after the first three trials following stimulus onset (Supplemental Fig. 1), which indicates that fish quickly adjusted and recalibrated their response when first engaging in the assay. Together, these results suggest that natural visual experience was not necessary for the emergence of the OMR, prompting the study of earlier mechanisms for the maturation of this complex behavior.

Even in the absence of sensory stimuli, spontaneous activity patterns, such as retinal waves, are thought to provide informative signals for the self-organization of neuronal circuits during development[17,51,52]. In the zebrafish visual system, such spontaneous activity in retinal ganglion cells (RGCs) begins at 2.5–3.5 dpf[53,54], right before RGCs form functional projections to the optic tectum (by 4 dpf)[55,56], illustrating a well-characterized synaptic pathway for visually-guided behaviors that could be subject to activity-dependent maturation processes. To test to what extent spontaneous neuronal activity, across any and all circuit elements, is necessary for the maturation of functional neuronal circuits in larval zebrafish, we developed a protocol to raise zebrafish under complete anesthesia (see Methods, Supplemental Fig. 2). Amongst a series of potential candidates that are used as anesthetics in fish and amphibia, we settled on tricaine, a sodium-channel blocker known to reversibly silence neuronal action potentials[57–60].

We first measured the acute effect of tricaine on spontaneous and visually-evoked swimming behavior (Fig. 2a, top), and observed that all animals stopped moving within 10 s of tricaine immersion, at which point fish generally lost postural control (Supplementary Movie 1). Further, we found that fish regained their ability to move within minutes of anesthesia washout (Supplementary Movie 2). By contrast, return to routine swimming behavior occurred with a time constant of 25 min (Fig. 2b).

To assess how thoroughly tricaine blocks spiking activity at the concentration used, we first performed in vivo whole-cell patch clamp recordings while tricaine was applied or washed out (Supplemental Fig. 3). We confirmed that evoked and spontaneous action potentials, as well as spontaneous excitatory postsynaptic potentials (EPSPs) were reduced within minutes and largely eliminated within hours after tricaine application.

In order to confirm tricaine's function as a global and reversible blocker of neuronal activity, we next performed functional calcium imaging experiments to quantify brain-wide responsiveness under anesthesia (Fig. 2a, bottom). We note that intracellular neuronal calcium signals have become an established and reliable proxy for sodium spikes[61]. Thus, quantitative analysis of calcium dynamics allowed us to

obtain a global readout of the effectiveness of tricaine's ability to block action potentials. We began by imaging normally-reared zebrafish (1) before anesthesia, (2) during a one-hour tricaine treatment, and (3) after washout of the drug (Fig. 2a, d). In this way, we quantified the magnitude of tricaine's effect and the timescale of neural activity recovery after anesthetic washout.

We first measured the effect of tricaine on visually-evoked activity across several brain areas known to be involved in processing visual information, including the tectum, pretectum and anterior hindbrain (Fig. 2c). We found that anesthetic application dramatically reduced neuronal activity within minutes (Fig. 2d). We note that some residual activity remained in the tricaine period (arrowheads, Fig. 2d), which mostly corresponded to cells ramping down in activity (Supplemental

Fig. 4a) and is consistent with our electrophysiological recordings, which showed that sparse residual spiking might occur in the first hour of tricaine application (Supplemental Fig. 3).

After anesthetic washout, activity began to re-emerge within an hour, which is in general agreement with the observation that extended recovery times were necessary to restore fish to full behavioral vigor after similar treatment (Fig. 2b). While we found that the population of sampled cells exhibited significant heterogeneity in the time required for full recovery to baseline responsiveness (Fig. 2d, Supplemental Figs. 5–9, 18), it is likely that only a small fraction of these neurons were critically involved in producing basic swimming (Fig. 2b). Thus, behavioral recovery from anesthesia need not be precisely matched to the recovery of global neuronal activity.

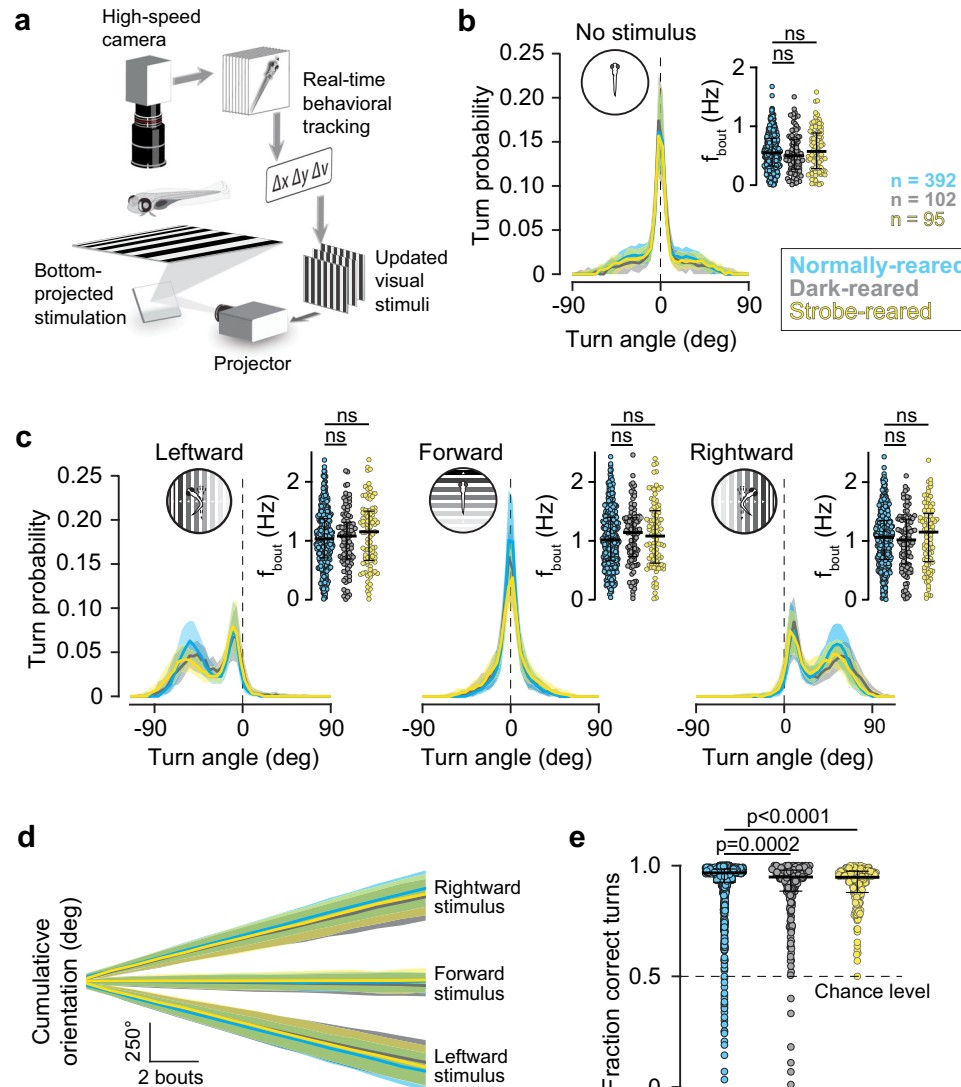

**Fig. 1 | The OMR matures in the absence of visual experience. a** Experimental setup; freely swimming fish are monitored with a camera while left- or rightward moving gratings are presented. Heading angle (Δv) and fish position (Δx, Δy) are extracted to lock visual stimuli to the animal's body axis. Modified with permission from Naumann et al. (2016). **b** Turn statistics of fish in absence of visual stimulus. Lines indicate median probability of swim bout angles over 30 trials for fish reared under a normal 14/10 h light/dark cycle (blue), in total darkness (black), and for fish reared under a 14/10 h cycle consisting of a 1 Hz strobe light and darkness, respectively (yellow). Error bands indicate quartiles of the probability of performing a given swim bout. Inset, average bout frequency of individual fish under the same experimental conditions; dots, bout frequency of individual animals across all trials; horizontal bar, median; error bars, interquartile range; n, number of animals (same across all panels). **c** Turn statistics and bout frequencies of fish when shown leftward-moving (left), forward-moving (center), and rightward-moving (right) gratings. Panel layout as in (**b**). **d** Cumulative change of the animals' rotation over time for control, dark-reared and strobe-reared fish. Lines and error bands indicate median responses and quartiles for rightward-moving (top), forward-moving (middle), and leftward moving stimuli (bottom). **e** Proportion of "correct" turns made by control fish, dark-reared fish, and strobe-reared fish over 30 trials. Dots, performance of individual animals across all trials; horizontal bar, median; error bars, interquartile range. The non-parametric, two-sided Kolmogorov-Smirnov test was used to test for significant differences in all panels. Source data are provided as a Source Data file.

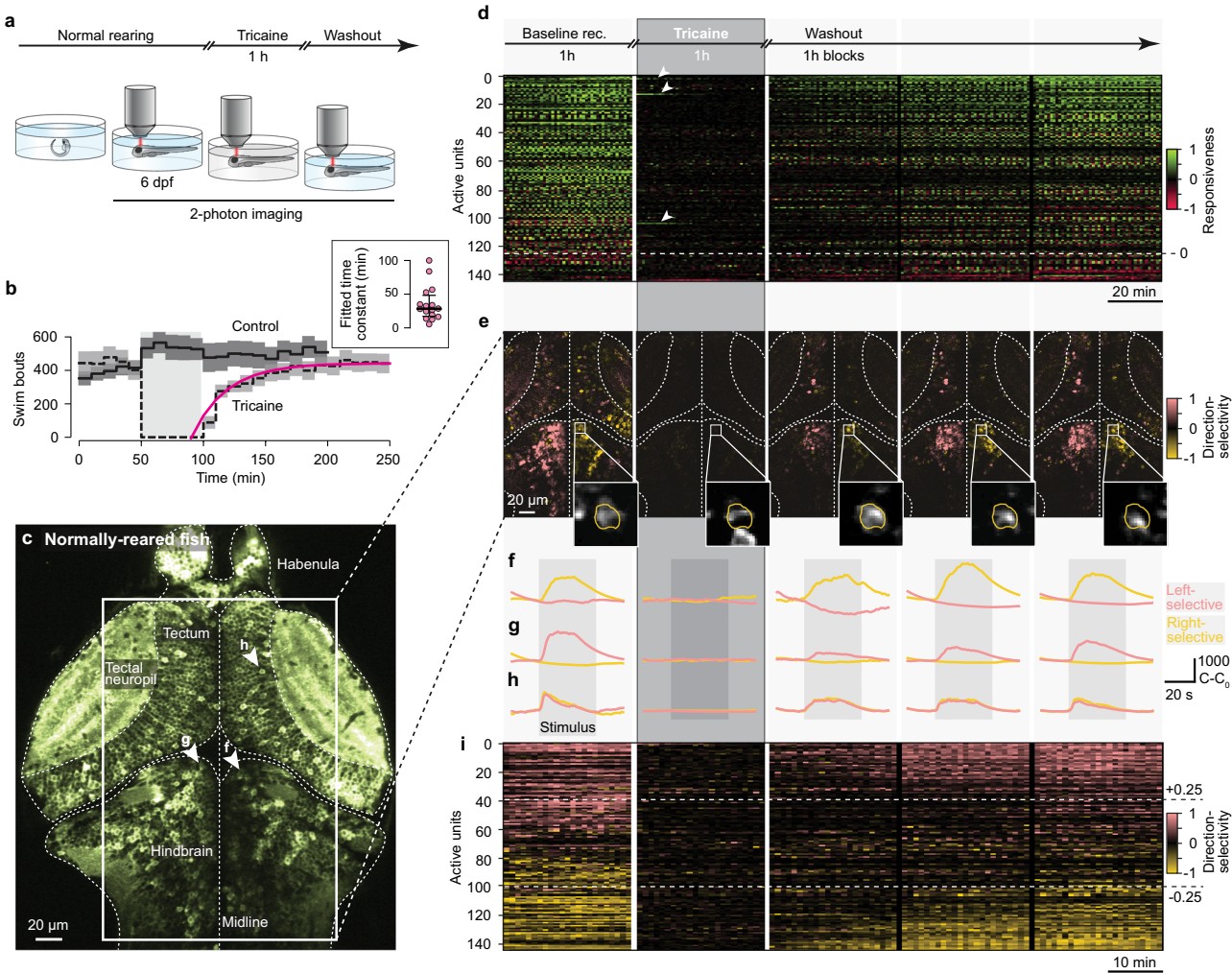

**Fig. 2 | Tricaine anesthesia reversibly silences neuronal activity. a** Experimental paradigm: normally-reared fish are imaged at 6 dpf before, during, and after acute, one-hour tricaine anesthesia. **b** Effect of acute tricaine anesthesia on the number of bouts performed during OMR. Dashed black line, performance of fish that were anesthetized for one hour (in shaded region, $n = 13$); solid black line, control animals that were not anesthetized ($n = 15$). Lines, mean bout rate; bands, standard deviation; pink line, exponential fit of recovery of bout rate from anesthesia. Inset, time constants of recovery across animals ($n = 16$); horizontal bar, median; error bars, interquartile range. **c** Cross-section of imaged brain prior to tricaine administration, regions involved in visuo-motor processing are indicated. Box, cross-section analyzed in (**d–i**); arrowheads, individual units shown in f-h. Cross-sections of all imaged animals are shown in Supplemental Figure 12. **d** Top, schematic showing time-course of imaging experiment, each block contains 1 h of recording. The remainder of the figure is aligned to these imaging blocks. Bottom, responsiveness-index of units in tectum, pretectum, and hindbrain ($n = 3$ fish) across trials. Units above dashed line showed increased responsiveness to visual stimulation, units below showed decreased responsiveness. Arrowheads; most active units during anesthesia (traces shown in Supplemental Fig. 4a). **e** Leftward (salmon) and rightward (gold) direction-selectivity index computed for cross-section in box in c. Insets, rightward-selective example unit (yellow outline) that was imaged across the entire experiment (corresponding traces in f; each image normalized separately). **f–h** Averaged trials showing stimulus-evoked activity of rightward-selective (**f**), leftward-selective (**g**), and motion-selective (**h**) units. Note that visual responsiveness disappears during anesthesia and the same tuning re-emerges during washout. **i** Direction selectivity-index of units in tectum, pre-tectum, and hindbrain ($n = 3$ fish) across trials. Units above top dashed line, left-ward-selective; units below bottom dashed line, rightward-selective. Note that relatively noisier signal during anesthesia is caused by unit-wise normalization used to compute direction-selectivity index (see Methods). Units in (**d, e, i**) consist of 1–5 neurons with the same response properties and thus represent small computational circuit blocks. Source data are provided as a Source Data file.

We next extended our analysis to neuronal response types that are critical for the OMR. Globally, we observed distinct left-selective and right-selective neuronal populations (Fig. 2e, salmon and gold, respectively), which became unresponsive under tricaine application and gradually returned to baseline responsiveness levels in the hours after anesthesia washout. To quantify the extent of anesthesia and recovery, we identified individual right-selective, left-selective and motion-selective cells that we were able to record from for the entire duration of the experiment (Fig. 2f–h, Supplemental Figures 5–9, 18). We found that similar to global activity patterns, the specific response properties of individual cells disappeared with the onset of anesthesia and returned to baseline gradually after the removal of the block

(Fig. 2f–h). Figure 2i highlights this phenomenon by showing the direction-selectivity of many units over individual trials throughout the five hours of the experiment. Here, it again became apparent that anesthetic application dramatically reduced neuronal activity from baseline already by the first visual-stimulation trial, and that neuronal activity gradually recovered after washout with comparable dynamics to the emergence of directional responsiveness.

Having validated the efficacy of tricaine as a reliable and reversible anesthetic for the larval zebrafish brain, we next utilized the drug to block all neuronal activity from 36 h post-fertilization until behavioral testing at 6 dpf. This four-day period of neuronal inactivation includes all critical phases of circuit maturation and starts well before the

emergence of spontaneous neuronal activity in larval zebrafish at 2–3 dpf[53–56]. We first performed behavioral testing immediately after anesthetic washout, moving fish directly from the anesthetic bath into the experimental rig. Then, informed by the multi-hour recovery of neuronal activity observed after the one-hour tricaine exposure (Fig. 2, Supplemental Figs. 5–10), we quantified behavior after two, six, and 24 h after release from anesthesia, during which fish were placed in anesthetic-free water in the dark, thereby minimizing visual experience during recovery.

A lifelong block of neural activity may introduce several adverse effects that we aimed to test: (1) Fish might not be able to swim at all, because the fine coordination of the central pattern generators (CPGs) in the hindbrain and spinal cord might require developmental activity to mature[62–64]. (2) Fish might not be able to see, because, equivalent to kittens that are raised with binocular lid closure[65], structured activity in visual centers might be required to connect the eyes properly to the central brain. (3) Fish might not be able to distinguish left- from rightward motion because the emergence of direction selective units requires developmental activity[66,67]. (4) Fish might not be able to accumulate visual motion and integrate information during the first few seconds of the stimulus because the integrator circuits in the hindbrain might require activity for accurate circuit assembly[37,38].

We found that even immediately after tricaine removal, all anesthesia-reared fish could swim, as indicated by a non-zero bout rate under all stimulus conditions (Fig. 3b) and they could see, as apparent from an increase in bout rate when presented with forward-moving gratings (Fig. 3b). Despite this clear indication that tricaine-reared fish respond to visual stimuli, we note that their swimming frequency was significantly disrupted by the treatment, never exceeding 50% of control animals, even after extended anesthetic washout. While anatomical defects of the body likely contributed to this reduction (Fig. 3a), it is also possible that the absence of developmental activity has a significant influence on the maturation of the CPGs that presumably control the bout clock in larval zebrafish.

Next, we tested the ability of tricaine-reared fish to perform OMR. We found that treated animals started to compute the direction of whole-field motion immediately after washout, as their turning probability was immediately biased towards the "correct" stimulus direction (Fig. 3c, 0–1 h, median of 55% accuracy of turns). Over the subsequent periods, performance consistently increased, reaching 75% median accuracy by 2 h of washout, 89% median accuracy by 6 h, and 93% by 24 h, comparable to the 96% observed in normally-reared fish (Fig. 3c).

An important feature of the OMR is that animals accumulate visual evidence within the initial period of stimulus exposure and are therefore able to improve their performance over the first series of bouts[37,38]. To test whether and when this ability to integrate information emerges after tricaine washout, we analyzed performance within the first few seconds after the onset of the stimulus. In agreement with recent studies[37], we found that all animals improved their performance over consecutive bouts within the initial time period, with integration time constants of several seconds (Fig. 3e). This was apparent as early as two to three hours after washout, and, critically, it matched the recovery of neural activity in animals anesthetized for one hour only, as described in Fig. 2. Notably, a significant fraction of direction-selective cells showed qualitatively similar integration time constants (Fig. 2f–h), neurons which are likely causally related to the within-trial improvement in performance.

We next tested the effects of shorter tricaine-treatment periods on behavioral performance. To that end, we anesthetized fish for several possible one-, two-, or three-day periods during the first six days of their development (Fig. 3d). Most treatment cohorts were allowed to recover for at least one day or more from the anesthetic before behavioral testing (Fig. 3d, gray), with the exception of four groups, in which fish were left in the anesthetic until the time of

behavioral testing and recovery was limited to two hours (Fig. 3d, pink). While fish undergoing four-day tricaine treatment required six hours of recovery to reach full behavioral performance, we found that shorter periods of anesthesia had only marginal effects on OMR performance: no single-day, two-day or three-day treatment led to a drop below 90% accuracy, even when measured after a recovery period of only 2 h (Fig. 3d). Complete recovery curves after these respective treatments, which showed performance at zero, two and six hours after washout, reinforced the notion that shorter tricaine exposures had very little effect on OMR performance (Supplemental Fig. 11).

One explanation for the extended, six-hour recovery time that was only apparent after four days of anesthesia could be a compromised general physiology caused by the metabolic insult of the extended tricaine exposure itself, which included uninflated swim bladders and hunched backs (Fig. 3a). These fish were not only deprived of spiking activity throughout development, they also never swam or utilized their muscles until washout, which in itself has been shown to cause decreased cell proliferation in the larval zebrafish forebrain[68]. In order to extract distinct anatomical phenotypes as a consequence of prolonged tricaine exposure, we examined high resolution 3D volumes of four control fish alongside four tricaine-raised animals (Supplementary Movie 3, Supplemental Fig. 12). While careful inspection of all cross-sections and brain regions indicated extended ventricles and sparse apoptotic cells from tricaine treatment (Fig. 4b), a quantitative analysis revealed that cell densities in all tested brain regions were not significantly different between control and anesthetic-reared animals (Fig. 4j).

An alternative explanation for the extended recovery time after our four-day anesthetic protocol is that functional neuronal circuit development requires several hours of activity-dependent tuning. That is, fish reared under anesthesia might not have an innate ability to perform OMR, but rather utilize the tricaine-free period in order to perform rapid, activity-dependent rewiring of their visual circuits. To distinguish between these two alternatives, we next set out to perform brain-wide functional imaging experiments in animals that emerged for the first time in their lives from tricaine anesthesia (Fig. 4a).

As before, we first tested the drug's efficacy when applied for extended time periods by performing in vivo patch-clamp recordings in animals that had been exposed to tricaine for many hours. Reassuringly, we found that for all extended treatment durations, spontaneous action potentials were eliminated and the frequency of EPSPs was reduced to levels that likely reflect spontaneous release events (Supplemental Fig. 3).

Next, we measured visually-evoked activity across brain areas known to be involved in processing visual information (Fig. 4b), (1) during continued tricaine treatment and (2) after washout of the drug (Fig. 4a, d). Reassuringly, during the first hour of recording (while animals remained anesthetized), no responsiveness to visual stimulation could be seen (Fig. 4d, leftmost panel), thus confirming a complete block of activity by tricaine anesthesia (Supplementary Movie 4). Importantly, residual fluorescent signals in the tricaine period (Fig. 4d, arrowheads) could be explained by noisy background fluorescence that gets amplified by our normalization procedure and show no correlation with the visual stimulus (see Methods, Supplemental Fig. 4b). Next, we used the emergence of activity during the period after washout to identify individual neuronal response types, such as motion- and direction-selective neurons, whose role is well-characterized in the OMR circuit[33] (Fig. 2). It is possible that these cell types might first become active in an unspecific fashion and only acquire their tuning properties gradually, which would indicate a need for activity-dependent mechanisms throughout development. Alternatively, cell types might appear immediately after washout in a fully-tuned fashion, which would suggest that the network developed into a functionally-tuned state in the complete absence of neuronal activity and just needs to be relieved from anesthesia.

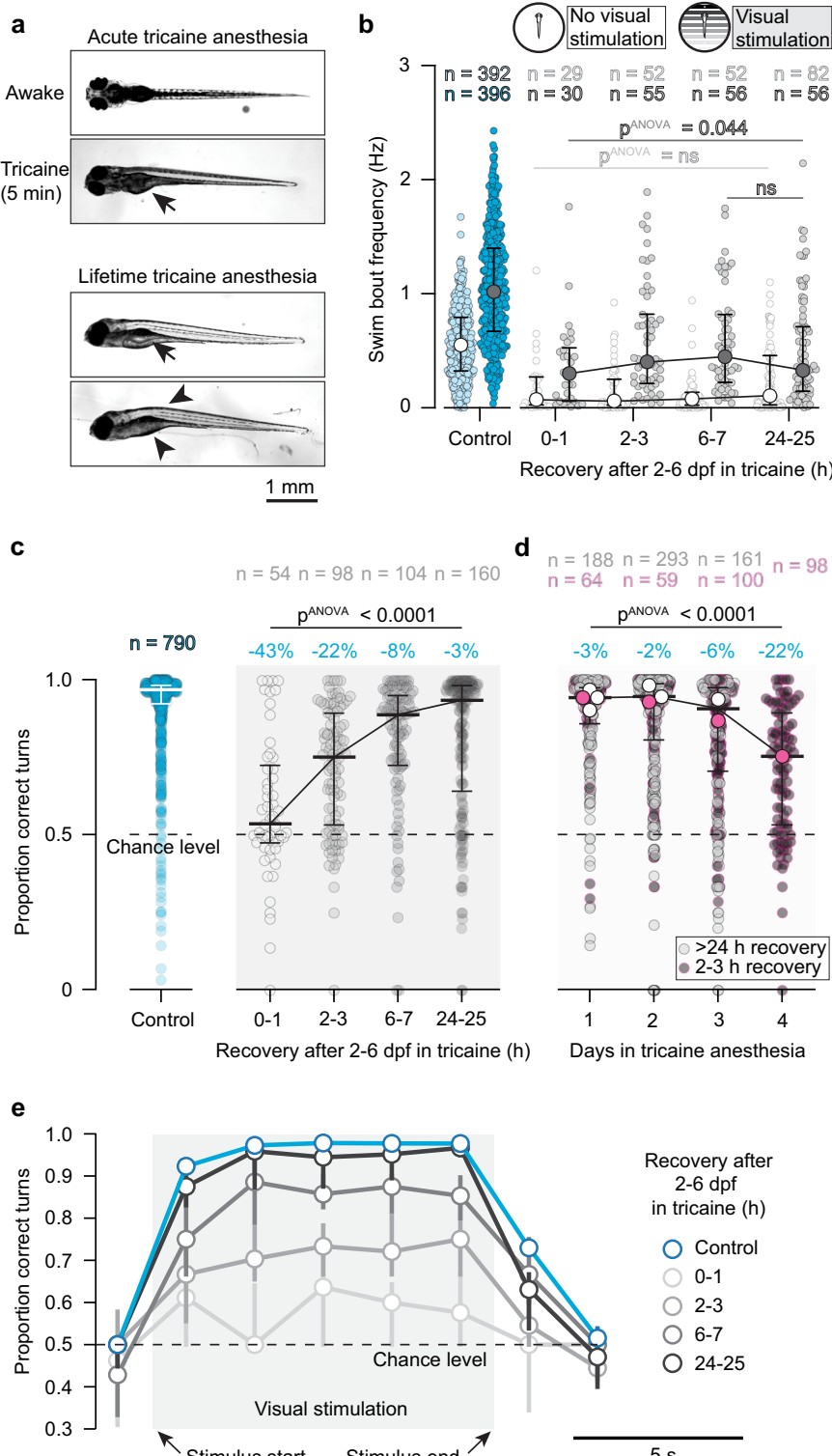

Remarkably, we found that in animals who were anesthetized for the entire duration of development, the responsiveness to visual stimulation and the direction-selectivity tuning of the visual circuitry emerged from anesthesia at equivalent rates and with similar delays (Fig. 4d, e, i, k; Supplemental Fig. 13b). Crucially, this indicates that cells do not gradually refine their direction selectivity over the course of the recovery period, but rather that they exhibit mature tuning as soon as they become active.

We note that stimulus responsiveness emerged with a significantly shorter delay in animals that were treated only for one hour

(mean onset of 45 min rather than 90 min in the lifetime cohort, Fig. 4k). Yet, more importantly, these animals displayed the same remarkable synchrony between the re-appearance of responsiveness and orientation tuning after anesthetic washout as observed in tricaine-reared individuals. The most parsimonious explanation for the delayed recovery associated with extended tricaine treatment is that the anesthetic may have to be cleared from the system for neuronal activity to recover.

Finally, we focused our analysis on the relatively slow stimulus-response time constants that we observed in many direction-selective

**Fig. 3 | Tricaine-reared animals can see, swim, and integrate visual stimuli.**
**a** Top, top-view of awake and acutely anesthetized zebrafish, showing loss of postural control immediately during anesthesia (Supplementary Movie 1); arrow, inflated swim bladder. Bottom, top-view of representative tricaine-reared animals showing loss of postural control and adverse morphological effects, including un-inflated swim-bladder (arrow), hunched backs, and abnormal abdominal morphology (arrowheads). **b** Comparison of swim bout frequency of control fish (blue) and tricaine-reared fish (grey) for different recovery durations after anesthetic washout. Light shading, no visual stimulation; dark shading, visual stimulation; dots, medians; error bars, quartile range; pANOVA: non-parametric Kruskal-Wallis test; 6–7 and 24–25 h (during visual stimulation) were compared with the two-sided, non-parametric Kolmogorov-Smirnov test). **c** Proportion of "correct" turns during OMR by control fish (blue, left) and tricaine-reared fish (grey, right) who were tested after different recovery durations after washout. Dots, averaged performance per animal; horizontal bars: medians; error bars, quartile range; blue percentages; effect sizes computed with two-sided Mann–Whitney test as absolute differences of median compared to control (all $p$-values < 0.0001); pANOVA: non-parametric Kruskal–Wallis test. **d** Proportion of "correct" turns during OMR by fish raised under anesthesia for different durations. Small dots, averaged performance per animal; horizontal bars, medians across animals; error bars, quartile range. Large dots, medians for each condition (e.g., "1 day" includes fish raised in tricaine for 24 h starting on either days 2, 3, 4, or 5 of development (i.e., 4 groups), "2 day" includes fish anesthetized for any 48 h-period (3 groups), etc.). All fish taken out of anesthesia on the day of testing received 2 h of washout (pink circles), therefore "4 days" group in (**d**) corresponds to "2-3 h" group in (**c**); all other fish recovered for >24 h before testing. Blue percentages, effect sizes computed with two-sided Mann–Whitney test, as in (**2c**) (all $p$-values < 0.0001); pANOVA: non-parametric Kruskal-Wallis test. **e** OMR performance as a function of time during visual stimulation. Dots, median performance in time bins; error bars, quartile range; same n as in (**c**). Source data are provided as a Source Data file.

neurons. This phenomenon was discovered in previous studies and uncovers a core computational feature of neuronal circuits underlying evidence accumulation, information integration and decision-making in the context of OMR-like stimuli[37,38,69]. In agreement with these studies, we confirmed that, in normally-reared animals, the integration time constants in direction-selective neurons following stimulus onset were distributed across a wide range (0.2 s to 3 s, Fig. 4l, black), while the time decay constants after stimulus offset lasted up to 20 s (Fig. 4m, black). We found that similar distributions of time constants reemerged after one hour tricaine exposure (Fig. 4l–m, gray). Critically, the distributions of time constants in direction-selective units that emerged for the first time after life-long tricaine anesthesia matched those observed after a single hour in tricaine (Fig. 4l–m, pink). This indicates that the ability to integrate and retain information over many seconds was already present in animals reared under anesthesia, and that this higher computational feature therefore does not require developmental spiking activity to manifest itself.

Together, these findings indicate that the emergence of functionally-tuned circuits in tricaine-reared fish does not depend on neuronal activity-based rewiring during washout. Rather, we suggest that functional response types are established by spiking activity-independent mechanisms and that the washout period serves solely to remove the damping effects of anesthesia.

In summary, our results suggest that the emergence of the neuronal circuitry for seeing, swimming, and sensory-motor integration is remarkably independent of developmental neuronal activity. Instead, we propose that many of the computational building blocks of the underlying functional circuitry, including responsiveness to visual stimuli, direction-selectivity, motor command neurons and CPGs in the spinal cord develop in an activity-independent manner under the exclusive control of genetic, transcriptional and cellular signaling algorithms emergent from the genome of the animal[70,71].

## Discussion

Here, we have developed a reversible developmental block of sensory stimuli and neuronal activity in a vertebrate that, when removed, reveals a functional and appropriately-tuned brain. In this way, we were able to separate "nature" and "nurture" for the first six days of an animal's life, thereby providing a toolkit for studying the extent of "innate" neuronal wiring in a vertebrate model system. Using this approach, we demonstrate the remarkable degree to which activity-independent developmental programs precisely pattern neuronal circuits.

We believe that our results contextualize, rather than overturn, a long literature of developmental perturbations, including eye sutures, sensory deprivations, and genetic silencing of activity that have suggested a critical contribution of neuronal activity for proper brain development. We note that these experiments generally relied on regional (and not global) plasticity perturbations and, crucially, they introduced a competitive imbalance between different modalities or mixed input channels. Amongst the most prominent examples are the monocular occlusion experiments by Hubel and Wiesel, where a clear competitive advantage was introduced to the eye that was allowed to remain open[9,22]. Interestingly, in less popular work, Hubel and Wiesel sutured both eyes shut prior to eye-opening, and nevertheless found the visual cortex in a remarkably unperturbed state[65], a result recently reproduced in monkeys[72]. Even the remaining aberrations of visual cortex neurons can be explained by the fact that perturbed visual inputs are outcompeted by other, non-visual modalities, such as motor-related or auditory signals, which have been demonstrated to contribute to primary visual cortex processing[23,24]. Another impressive example of competitive takeover of underused cortical regions is the finding that visual pathways can successfully innervate auditory cortex in ferrets when the afferent thalamic relay nuclei are surgically removed[25,73]. This phenomenon is not restricted to the sensory cortex: in the mouse olfactory bulb, it has been shown that blocking an individual olfactory channel leads to significant rearrangement of synaptic connectivity, while blockage of all receptors leaves the circuitry largely unchanged[74]. A classic demonstration of axonal competition at play is the emergence of ocular dominance columns in the otherwise monocular amphibian tectum if a third eye is artificially implanted early in development[75]. This phenomenon, which was shown to require spiking activity[76], was later confirmed in the zebrafish tectum, where selective pruning phenotypes in innervating RGC axons were observed when individual neurites are genetically silenced[77].

However, in experiments where all spiking activity, including action potentials in innervating axons, was blocked in large regions of the cortex, selective targeting of neuronal projections was still observed[67,78]. These results suggest that spiking activity might be necessary in certain contexts – however, they cannot distinguish between an instructive or merely permissive role of activity in the shaping of circuits[79]. Further, the small effect size of these perturbations suggests that the importance of these activity patterns in the structuring of neuronal circuits is minor.

Another experimental perturbation that was found to interfere with circuit refinement is the local application of synaptic transmission blockers to interrupt afferent neuronal inputs. For example, it was found that N-methyl-D-aspartate (NMDA) receptor antagonists, when locally applied to the optic tectum of fish[77,80], frogs[81,82], and rats[83] prevented pruning of incoming axonal arborizations from RGCs. Effects of synaptic silencing are also uncovered by local curare application to the neuromuscular junction in mice, which has been shown to prevent the segregation of motor neuron axons onto individual myofibers[84,85]. Classically, these results support the notion that activity-dependent processes are necessary for the refinement of circuit structure. However, these results can not distinguish whether spike-timing-dependent plasticity or spike-independent synaptic signaling is responsible for the observed refinement of circuit structure[86].

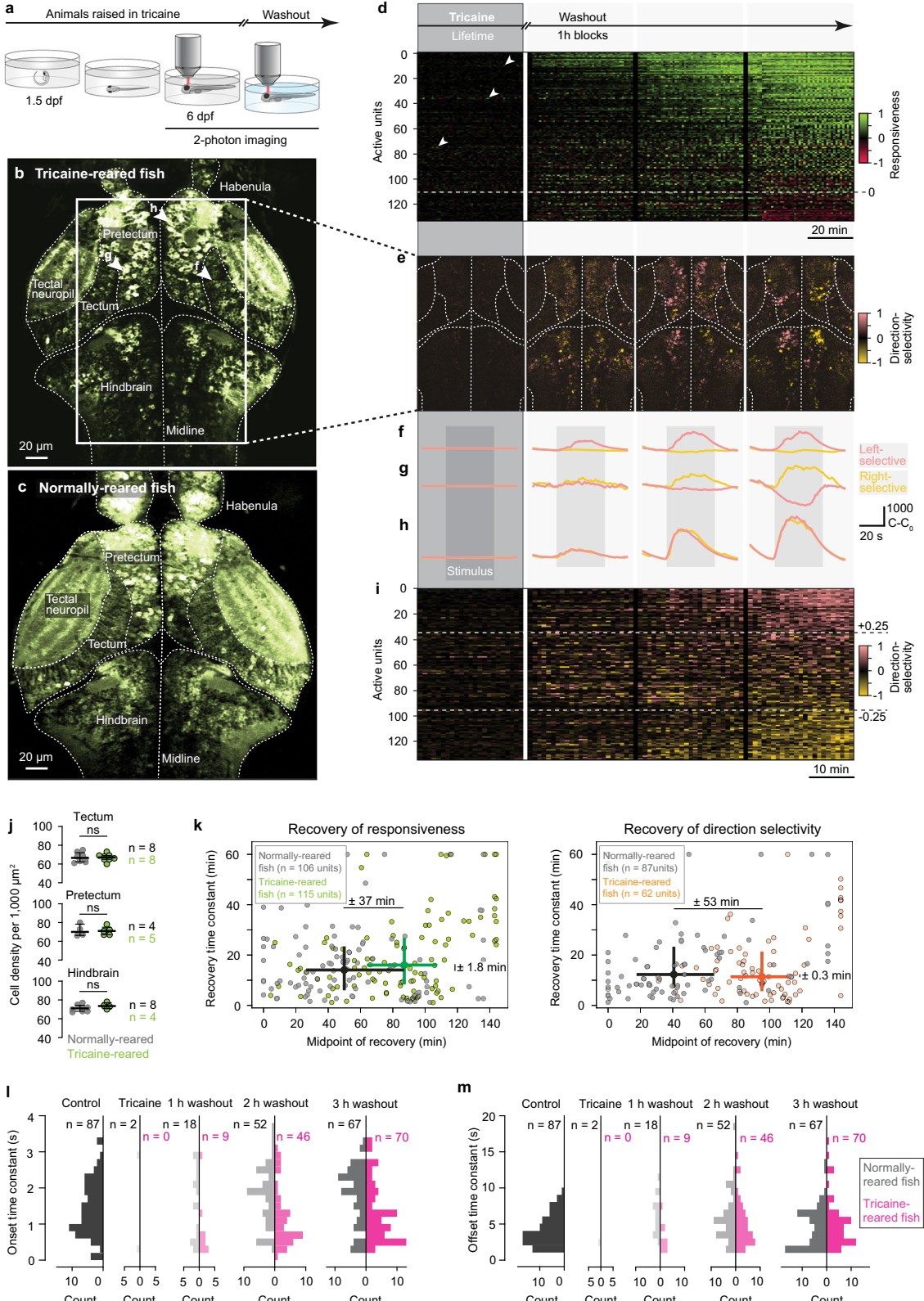

Specifically, the cited perturbations categorically block trans-synaptic, homeostatic signaling cascades, which can manifest in many different forms, including as calcium activity waves that have been observed to propagate across various brain areas in many different model organisms. Such communication pathways are also known to play a critical role in the signaling between pre- and postsynaptic elements in early development, contributing to synaptic target recognition and functional stabilization[87]. Our perturbation is limited to the block of "classical" sodium-driven action potentials, but does not prevent calcium dynamics unrelated to spikes, metabolic activity, or the spontaneous release of neurotransmitters, which are necessary for the fundamental exchange of information between pre- and postsynaptic elements. These processes, without which an organized emergence of brain circuits would not be possible, are therefore still operational and

**Fig. 4 | Visual circuits mature under silenced developmental activity.**
**a** Experimental paradigm: 6-dpf tricaine-reared fish are imaged before, during, and after washout. **b** Imaged cross-section of brain of tricaine-reared fish after washout, regions involved in visuo-motor processing indicated. Box, cross-section analyzed in (**e**); arrowheads, units in (**f**–**h**). Note extended ventricles and bright (putatively apoptotic) neurons. (Cross-sections for all animals in Supplemental Figure 12). **c** Cross-section for normally-reared, awake fish; similar plane as in (**b**) for comparison. **d** Top, schematic showing time-course of imaging sessions; the remainder of the figure is aligned to these imaging blocks. Bottom, responsiveness-index of units in tectum, pretectum, and hindbrain (n = 3 fish) across trials. Units above dashed line showed increased responsiveness to visual stimulation, units below showed decreased responsiveness. Arrowheads; most active units during anesthesia (traces in Supplemental Fig. 4b). **e** Leftward (salmon) and rightward (gold) direction-selectivity index for cross-section in box in (**c**). Note direction-selectivity emerges during washout. **f**–**h** Averaged trials showing stimulus-evoked activity of rightward-selective (**f**), leftward-selective (**g**), and motion-selective (**h**) units. **i** Direction selectivity-index of units in tectum, pretectum, and hindbrain (n = 3 fish) across trials. Units above top dashed line, leftward-selective; units below bottom dashed line, rightward-selective. Units in (**d**, **e**, **i**) consist of 1–5 neurons with same response properties and thus represent small computational units. **j** Cell densities across visual areas of tricaine-reared and control fish; n, number of hemispheres measured; horizontal bars, medians; error bars, quartile range; two-sided Kolmogorov-Smirnov test. **k** Comparison of recovery of responsiveness to visual stimulation (left) and of direction-selectivity (right) during tricaine washout between normally-reared and tricaine-reared animals (see Methods). Dots, individual units; crosses, median and quartile ranges. Significant effect sizes indicated (computed with Mann–Whitney test as absolute differences of medians between groups). **l** Distributions of onset time constants (when visual stimulation started) between normally-reared and tricaine-reared fish for different time-points after anesthetic washout. **m** Distributions of offset time constants (when visual stimulation ended) between normally-reared and tricaine-reared fish for different time-points after anesthetic washout. **l, m** n: number of fittable units (which exceed $C-C_0 > 100$ during imaging). Source data are provided as a Source Data file.

are most likely critically involved. It is therefore important to point out that our use of the term "neuronal activity" is restricted to sodium channel-dependent action potentials, which we hypothesize are not necessary for the emergence of structured circuits.

Together, most of these findings are compatible with the prediction that if all competition is removed by a categorical block of all neuronal spiking activity, as was done in our experiments, the "natural," genetically hard-wired brain structure should emerge, in spite of this rather drastic perturbation. Indeed, more than a hundred years ago, experiments in tadpoles already revealed that normal behavioral patterns can be observed in cloretone-reared animals, a treatment that was assumed to block all neuronal activity[14,15]. Later studies in zebrafish found normal emergence of tectal structures in fish where all sensory activity was prevented by the removal of the eyes[16].

The general observation that anatomically "normal" structures can robustly emerge in the absence of neural activity was confirmed in developing mouse embryos by showing that genetically blocking synaptic release in all neurons led to an anatomically unchanged macro-, micro-, and nano-structure throughout the brain[26]. Moreover, in newborn kittens, the binocular features of the visual cortex were shown to emerge even if eye-specific, monocular visual stimuli were explicitly decorrelated from each other[88], suggesting that the emergence of functional circuits in the cat visual system is governed largely by genetic programs. This was confirmed in a later study that explicitly identified a molecularly defined mechanism for the formation of ocular dominance columns in cats[89]. Together, these studies argue that functional neural circuits can emerge based primarily on genetic instructions, a conclusion that we have confirmed in the visual pathways of the larval zebrafish.

From an evolutionary perspective, a robust hardcoding of essential, basic behaviors is expected to be present in all animals. This can be seen in turtles heading out to sea after hatching[90], newborn zebras galloping alongside their herd, and freshly-hatched iguanas escaping from snakes[91]. Of course we acknowledge that, once the animal engages with the world, activity-dependent plasticity is obviously necessary to shape and update neural circuits, and to maintain them, which allows for calibration, tuning and maintenance of circuit structure, and the ingestion of new information. As such, we propose that a large fraction of adaptive motor sequences and sensorimotor reflexes may be a result of hard-wired and innate circuitry. By contrast, environmental changes that cannot be anticipated on evolutionary timescales require explicit learning and adaptation to develop novel behavioral modules[92]. Nevertheless, given that complex behaviors can certainly emerge from activity-independent developmental processes, open questions remain as to (1) which behaviors and circuits do require neuronal activity for maturation and (2) why activity is utilized for refinement of neuronal circuits, when developmental

processes could be sufficient. Further understanding and incorporating these neurodevelopmental processes may prove instructive to both computational neuroscience and machine learning communities in the future[93–95].

## Methods
### Zebrafish
For all behavioral experiments, we used 6 dpf wild-type (WIK) zebrafish. Fertilized eggs were collected in embryo water (Methylene Blue + E3 solution (5 mM NaCl, 0.17 mM KCl, 0.33 mM CaCl2, 0.33 mM MgSO4)) and transferred to filtered fish facility water after 24 h. Under standard rearing conditions, zebrafish were maintained on a 14 h light /10 h dark cycle at 28.5 °C. Dark-reared embryos were placed in a light-proof box after collection, ensuring a 24 h dark cycle. After collection, strobe-reared embryos were placed in a box that contained a 1 Hz strobe light, which was active during the 14 h light period (matched to the standard rearing condition's light on and off times) and off during the 10 h dark cycle. Immediately prior to testing, both strobe-reared and dark-reared fish were kept in darkness until placed in the experimental rig and the stimulation began. The fish were not fed under any conditions, as this would introduce an additional source of variation between standard rearing, where feeding is easier, dark/strobe, where feeding becomes more difficult, and anesthetized fish, where feeding is impossible.

All imaging experiments in this study were performed on the transgenic zebrafish line *elavl3:GCaMP6s* line[37]. We raised groups of 20–30 larvae in standard Petri dishes (90 mm diameter) containing filtered fish water on a 14 h light/10 h dark cycle at 28 °C. Imaging was performed at 6 dpf, at which age sex cannot be determined. Prior to imaging, we screened larvae for strong GCaMP6s fluorescence and the absence of any pigmentation. All experiments were approved by the Harvard University standing committee on the use of animals in research and training.

### Behavior experiments in freely swimming larval zebrafish
We performed free-swimming behavioral quantification with a previously described experimental setup[37]. Briefly, individual larvae were placed in a closed-loop virtual-reality environment and presented with a moving bar stimulus projected from below (60 Hz, AAXA P300 Pico Projector). In "no stimulus" conditions, only a gray background was shown. We used real-time behavioral tracking at 100 Hz to determine the location and body orientation of animals and updated the visual stimuli accordingly.

Each behavior trial began with a 5 s period in which gratings (spatial period of 1 cm and oriented along the animal's anterior-posterior axis) were static and locked to fish orientation. These orientation-locked gratings began moving at a velocity of 1 cm/s for

10 s, followed by another 5 s static period before the next trial was initiated. Each animal was presented with 30 sets of 'no stimulus' or forward-, backward, leftward-, or rightward-moving gratings. The order of stimuli presented within each set was randomized.

## Tricaine anesthesia

For all experiments, we used a concentration of 60–100 µg/mL tricaine in filtered fish water (MS-222 Sigma-Aldrich) buffered to pH 7-8. We came to this concentration through extensive testing of a wide range of concentrations. We optimized for a solution that kept fish fully anesthetized for the maximum treatment duration of 4 days, while minimizing physiological effects and mortality. We applied 60 µg/mL tricaine up until day 5, and utilized 80 µg/mL on day 5 and 100 µg/mL on day 6, raising the concentration to combat any indication of habituation to the solution. We note that different fish lines tolerate different levels of anesthesia, although all tested lines fell into the 60–100 µg/mL range. The tricaine bath was exchanged every 12 h because we found that tricaine started to lose its potency after around 18 h, presumably because it was metabolized by the animals kept in the dish (Supplemental Fig. 2). We assessed whether larvae were indeed fully anesthetized through shaking the plate and tapping fish, as well as a long-term quantification of movement response, or lack thereof, under a light-on and light-off stimulus (Supplemental Fig. 2).

We discarded all fish in a Petri dish if we found any animal that responded to visual or physical stimulus.

Zebrafish embryos were placed in petri dishes (9 cm diameter) filled with tricaine solution and kept in the dark until experimental testing, because tricaine forms toxic byproducts under extended exposure to light. Since the onset of spontaneous neuronal activity in larval zebrafish happens prior to hatching, we used a pronase solution (50 mg/ml in fish water for 5 min, Sigma-Aldrich) to dissolve the chorion prior to 24 h post fertilization (hpf). This allowed us to anesthetize animals at 36 hpf and to have animals develop fully in the absence of their ability to hatch themselves. The same pronase-treatment was also performed for all "tricaine-control" fish used in the study.

For tricaine washout, we transfered the fish to a petri dish filled with standard fish water, and placed the plate in darkness for the duration of washout, thereby preventing visually-guided "entrainment" of neural network during this period.

## Whole-cell in vivo patch-clamp recordings

To cross-check the efficacy of tricaine in blocking spikes, we performed in vivo whole-cell recordings from putative Purkinje cells in larval zebrafish. We chose to record from Purkinje cells because they integrate from a large number of parallel fiber inputs that carry a copy of all motor-related activity in the brain. This allowed us to not only monitor the efficacy of tricaine for blocking spikes in the recorded cells themselves, but it also enabled us to assess overall network activity representing a higher-order motor output system of the brain by measuring the rate of parallel-fiber EPSPs arriving at the somata of Purkinje cells.

Briefly, for tricaine wash-out experiments, 5–7 dpf larval zebrafish were anesthetized in fish water containing 100 µg/mL tricaine. After 30 min to several hours, anesthetized animals were embedded in 1.8% low melting-point agarose that was prepared with 100 µg/mL tricaine dissolved in artificial cerebrospinal fluid (ACSF), containing in mM: 134 NaCl, 2.9 KCl, 10 HEPES; 10 Glucose, 1.2 MgCl2, 2.1 CaCl2; the pH was set to 7.8 with NaOH. Following embedding, the agarose above the head of the animal was removed and the skin covering the cerebellum was carefully cut along the anterior-posterior axis with sharpened tungsten needles. Fish were immediately transferred to a custom-built electrophysiology setup where cerebellar neurons were visualized under an upright microscope (Olympus BX51) equipped with infrared differential-interference contrast and an 20X Olympus objective

(XLUMPLFLN). We identified putative Purkinje cells visually on the basis of their superficial location in the cerebellum and larger soma sizes. We established whole-cell recordings using pipettes pulled from thin-walled borosilicate glass (outer diameter: 1.5 mm, inner diameter: 1.17 mm) filled with intracellular solution containing in mM: 105 K-Gluconate, 16 KCl, 10 EGTA, 10 HEPES, 2 MgCl2, 4 Na-ATP, 1 Na-GTP, and 1 mg/ml Lucifer Yellow; the pH was set to 7.2 with KOH. The osmolarity of the intracellular solution was ~280 mOsm and the pipette resistance was ~ 8–10 MOhm. Recordings were sampled at 10 kHz, filtered at 3 kHz, digitized with a National Instruments board (PCIe-6323), and monitored and controlled using a custom-written LabVIEW package. Following break-ins, the bridge potential was compensated for and neurons were held at membrane potentials between −60 to −70 mV. First, we characterized the neurons' responses to subthreshold and suprathreshold current injections (Supplemental Fig. 3). We then replaced the tricaine-ACSF in the bath surrounding the fish with 15 ml of regular ACSF at a flow rate of 5 ml/min using an electrically grounded peristaltic pump system. Finally, we continued to monitor the cells' responses to the same current injections for as long as the recording remained stable. We confirmed with behavioral experiments in the same setup and using the same embedding techniques that animals regained their ability to perform evoked tail and pectoral fin responses after several minutes of the described wash-out procedure.

Tricaine wash-in experiments were performed in an analogous manner, with the exception that animals were paralyzed with 1 mg/ml bungarotoxin in fish water for 5–10 min prior to being embedded in 1.8% agarose prepared in ACSF. After establishing the whole-cell configuration and recording the neuronal responses to current injections, we then washed in ACSF containing 200 µg/mL tricaine. We chose a higher tricaine concentration for two reasons: 1) to account for the fact that the agarose surrounding the animal acts as a reservoir that retains regular ACSF and, 2) that recordings can become unstable due to the vibrations introduced by the peristaltic pump system, which necessitated brief wash-in periods using higher concentrations. We confirmed in a separate set of behavioral experiments in the same setup that fish lost their ability to perform evoked tail and pectoral fin responses after several minutes, which was comparable to free-swimming fish treated in fish water containing 100 µg/mL tricaine.

In each electrophysiological trace, we then counted the number of spontaneously occurring spikes and EPSPs during recording periods in which no current stimulation was performed, and the number of spikes that were evoked by the current injections. Traces with unstable membrane dynamics were excluded from the EPSP analysis.

## Two-photon calcium imaging

For in vivo imaging experiments, we prescreened larvae as described above. Larvae were then fully embedded in lukewarm agarose, which was allowed to solidify for one hour. We found that incorporating this resting period almost completely reduced drift of the imaging plane during recording sessions. Critically, this allowed us to record from the same identified units for extended periods of time.

Fish that were raised under tricaine anesthesia were embedded in an agarose solution containing 60 µg/ml tricaine to ensure that the embedding process would not relieve the anesthetic block. We used a custom-built two-photon microscope containing a femtosecond-pulsed MaiTai Ti:Sapphire laser (Spectra Physics) tuned to 950 nm for GCaMP6s imaging, a set of x/y-galvanometers (Cambridge Technology), a 20× objective (XLUMPLFLN, Olympus), and a photomultiplier that was amplified by a SR570 current preamplifier (Stanford Research). The setup was operated by a custom-written Python 3.7-based software package. We tuned the laser power at the specimen to ~13 mW and acquired images at frame rates of 1 Hz.

We imaged each plane at a spatial resolution of ~0.7 µm/pixel (700 ×700 pixels) for one hour, during which animals were presented with a bottom-projected visual stimulation paradigm. Each visual stimulation

trial lasted for 60 s and consisted of a 10-second period during which a stationary grating was presented, a 30-second period during which the grating then moved at constant velocity either to the left or the right of the animal (in a randomized fashion), and finally a 20-second period during which the grating was stationary again.

First, larvae were imaged for one hour under "baseline" conditions (i.e., in fish water or tricaine-fish water for acute or lifetime-tricaine experiments, respectively). During imaging sessions, the solutions surrounding the animal were static (i.e., not perfused). Then, we quickly removed the solution used during baseline imaging from the bath using a custom-built suction system (exchange rate: ~25 ml/min) and immediately filled the bath with the same volume of "experimental" solution (i.e., tricaine-fish water or fish water for acute or lifetime-tricaine experiments, respectively) using a custom-built gravity pump system (exchange rate: ~20 ml/min). Great care was taken to image from the same z-plane before and after the exchange of solutions, which necessitated small adjustments in the z-position of the objective during a brief "alignment period" (~1-2 min) before each new recording session was started. This process was repeated for each consecutive "washout" imaging-period (in fish water). Critically, this process enabled us to replace the solution surrounding the animal without opening the setup, changing the x-y position of the animal relative to the objective, or moving the objective by more than a few μm, and thus allowing us to image the same units for several hours under different anesthetic conditions (Figs. 2, 4).

### Preprocessing of two-photon imaging data

We processed our imaging data with the open-source toolbox CaImAn[96] and the Computational Morphometry Toolkit[97], as described before[37]. Briefly, we implemented a three-step processing pipeline, in which we first performed piecewise rigid motion correction using CaImAn (NoRMCorre; using standard parameters)[97,98].

For the single-unit analyses shown in Fig. 2f–h and Fig. 4f–h, we applied CaImAn's segmentation algorithm (CNMF; using standard parameters adjusted for frame rate and cell size). We used a relative measure for calcium dynamics by subtracting the baseline calcium level ($C_0$; taken in the 10 s period prior to visual motion onset) from the calcium activity during the entire trial (C). We used this $C - C_0$ metric instead of computing $\Delta C/C_0$, because $C_0$ was often zero. Importantly, for all single units analyzed in this paper, we confirmed visually that we indeed recorded from the same individual neurons during the entire duration of the experiment (i.e., a total of 5 h) and excluded units that moved in or out of the imaging plane during the experiments.

For the multi-unit analyses shown in Fig. 2d, i (147 units) and Fig. 4d, i (134 units), we hand-segmented small regions containing multiple neurons in motion-corrected image stacks. These were selected such that they contained clusters of 1–5 neurons that displayed the same functional responses during visual stimulation. Thus, these units correspond to small, similarly-tuned circuit motives across the tectum, pre-tectum, and hindbrain.

The responsiveness index was calculated by computing the average $C-C_0$ value for each trial and by normalizing these values across all trials for each unit individually. Importantly, by normalizing each unit to its own maximum $C-C_0$ value in this manner (as opposed to a global normalization), we sought to highlight subtle responsiveness levels, particularly during tricaine periods, as an additional confirmation that brain activity was indeed silenced by anesthesia. Units that showed a net increase in responsiveness to the visual stimulation were normalized to 1, while units with a net decrease in responsiveness during visual stimulation were normalized to −1. This allowed us to dissect how tricaine anesthesia influenced excitatory and inhibitory circuit properties, respectively.

To compute the direction selectivity index, we matched subsequent pairs of trials in which gratings were first moving leftwards, followed by rightwards. We then calculated the difference in fluorescence responses to these two stimuli, which, after normalization, gave us a metric that ranged from −1 (i.e., rightward-selective) to +1 (i.e., leftward-selective). Given that we normalized each cell's activity, we encountered a higher level of noise in the tricaine period, as low neural activity becomes indistinguishable from a lack of direction selectivity.

Midpoint ($x_0$) and time constant (k) of recovery in Fig. 4k were obtained from fits with logistic curves (see Supplemental Figs. 5–9, 14–19).

### Statistics and Reproduciblity

For all studies quantifying OMR behavior we ran at least N = 64 fish, significantly above the N = 30 recommended by papers that utilized the same behavioral rig[37], although exclusions applied that lowered these numbers in certain conditions. For imaging, we ran at least 7 fish per condition, and utilized the three of the brightest samples per condition. For the embedded imaging experiments (Figs. 2, 4), variability between fish was low, and hence, we planned experiments for around N = 3 fish. We then performed experiments in batches according to these numbers.

For freely swimming fish, we excluded animals when they did not swim at all or spent most of the time near the wall of the experimental chamber, and therefore could not be tracked. For anesthesia-reared fish, we discarded all fish in a petri dish if we found any animal that responded to visual or physical stimulus. We excluded any day of behavioral recording when the wild type (untreated fish) showed significant deviations from normal behavior. We excluded fish with weak GCaMP activity from neural recordings.

Data for each condition was collected on separate days and over multiple months, and from multiple parent batches, all with consistent results. For behavioral data, this represented daily testing of WT fish, and at least a few days of repeats of all other treatments. 2-photon imaging studies were performed over the course of 2 months, providing 3 replicates per control and treated condition, with additional replicates eliminated when GCaMP activity was low.

Fish were randomly assigned to treated, untreated, and recovery conditions. The order of visual stimuli was always presented randomly.

All data analysis was automatic, and hence blind, and was not different between different conditions within one experiment. Investigators were not blinded to group allocation during data collection, however, most, if not all, data collection occurred using automated behavioral experiments.

### Reporting summary

Further information on research design is available in the Nature Portfolio Reporting Summary linked to this article.

## Data availability

Source data are provided with this paper. The raw imaging data have file sizes on the order of terabytes, and can be shipped on a hard drive at request from the corresponding author. Source data are provided with this paper.

## Code availability

The code used to generate figures is available at https://doi.org/10.5281/zenodo.10262467.

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

## Acknowledgements

We thank Andrew Bolton for helping conceive the study and for assistance in pilot experiments. This work was supported by NIH NIGMS T32 GM008313 Grant (D.L.B.), by Swiss National Science Foundation Postdoc Mobility Fellowships P2EZP3_188017 (G.F.P.S.) and P500PB_203130 (G.F.P.S.), and by NIH Grant U19NS104653 (F.E.), NIH Grant 1R01NS124017 (F.E.), NSF Grant IIS-1912293 (F.E.), and Simons Foundation SCGB 542973 (F.E.).

## Author contributions

D.L.B. and F.E. conceived the project. D.L.B. developed the anesthetic protocol, performed behavioral experiments, and analyzed resultant data. G.F.P.S. performed in vivo patch-clamp experiments and two-photon imaging experiments and analyzed electrophysiological data. D.L.B. and G.F.P.S. analyzed imaging data. All authors contributed to preparing figures and writing the manuscript.

## Competing interests

The authors declare no competing interests.
