## [Peer Review File · Nature Communications]

REVIEWER COMMENTS

Reviewer #1 (Remarks to the Author):

The paper by Barabási and colleagues asks to which degree the development of circuits and behaviors in the zebrafish brain depends on neuronal activity, be it sensory evoked or spontaneous. They start by showing that a simple, visually driven behavior, the optomotor response (OMR), is present in larvae that were completely deprived of vision throughout development. Importantly, even during the very first testing trials after dark rearing, fish largely showed correct responses, indicating that they were able to perform this behavior without any "training". Next, in order to probe for a potential role of spontaneously generated activity in establishing the behavior, they raised zebrafish under tricaine, which, as they show using calcium imaging, abolishes all neuronal activity. Despite this, fish showed correct OMR behavior after only a very short period of drug washout. Tricaine raised fish eventually matched control levels, however, only after an extended period of drug washout. Finally, using calcium imaging in key brain regions involved in the OMR, they show that very shortly after drug washout, neurons show tuning for stimulus direction and other, more complex features, like evidence accumulation. The authors conclude that the circuits underlying these computations are assembled without neuronal spiking activity.

These are exciting results, which make a strong case in the old debate on the role of activity in the formation of neuronal circuits. While the paper very certainly won't close the case, it provides compelling evidence that, at least for the system used, neuronal activity is far less important than many researchers would have thought.

I have a few questions and comments, mostly minor, which might help improving the manuscript.

1. I am missing statistical tests for some of the data reported.
2. In the introduction, the authors are slightly loose on the development of ocular dominance. "Binocular rivalry" is not exactly what drives this process, but rather describes the psychophysical phenomenon of rapidly alternating perceptual dominance of one or the other eye. Also, do they mean "ocular dominance formation" or formation of ocular dominance columns? The citations given here (9-11) are not exactly the correct ones. References 12 and 13 did not report structural abnormalities.
3. Authors should clarify for how long dark reared fish were exposed to light (if at all) before going into the first behavioral trials. I am asking, since many experiments have shown that even brief exposure to stimuli can have effects on the development of behavior and response properties.
4. Figure 2e, and others: colors are referred to as yellow and pink, and then salmon and gold elsewhere.
5. Methods: "simulation" should read "stimulation".
6. Fig. 4k: sure about the numbers for one hour and lifetime tricaine? Seems they are not matching between Figure and main text.
7. They state "Careful inspection of all cross sections and brain regions, under active and silenced conditions, did not reveal any clear differences." But that seems to be at odds with other statements (e.g. Fig. 4b legend).
8. In several instances, mostly in the supplemental Figures, axis labels are missing or unclear, e.g. S Fig. 16.

Reviewer #2 (Remarks to the Author):

This manuscript uses a sodium-channel blocker to abolish the visible neuronal activity using calcium imaging during development of larval zebrafish. The goal is to differentiate between hardwired circuits (nature), and the need for sensory stimuli (nurture) to develop a proper OMR response. The manuscript is well-written and clear, the results support most of the claims. However, the extensive existing literature on the topic makes the results unsurprising, and this work merely incremental.

There have been many studies looking at the development of visual-driven behaviors in zebrafish (and many animal models), which are not discussed in enough detail in the introduction. Enucleation, dark rearing, lesion, or other approaches have been used for decades now, and it is unclear what this manuscript offers beyond the use of anesthetics for part of the development. The concept of "hardwired" genetically encoded modules has a long history, which the authors only mention at the end of their results and in the discussion, but should be present in the introduction. A lot of the existing literature is only mentioned in the discussion, and the studies of the Sumbre and Goodhill labs on early zebrafish visual development in the absence of stimuli are only mentioned in passing.

The use of tricaine is intriguing, but the use of calcium imaging could hide subthreshold activity, or sparse firing resulting in low signals lost in the noise, which could still be enough spontaneous activity for the circuits to develop on top of genetic information. It is also unclear what effect, if any, tricaine would have on the retinal circuits, which are essential for direction selectivity, and whose retinal waves have been reported to not depend on action potentials (see 10.1016/S0896-6273(00)81121-6). So, some of the tuning observed could still be activity dependent, but in the retinal circuits. Electrophysiology would be necessary to assess how thoroughly tricaine blocks neuronal activity at the concentration used. Furthermore, the authors themselves acknowledge the side effects of the long-term anesthetic use, making any inference difficult with the data available.

Reviewer #3 (Remarks to the Author):

In this manuscript the authors describe their experiments, using zebrafish, that test the role of nerve cell activity in the development of a specific circuit. The authors chose to examine optomotor responses (OMRs) as this is a behavior that can be quantified and is due to specific cells in the pretectum, tectum, and hindbrain. The authors compare responses of zebrafish larvae raised in total darkness, larvae subjected to strobe lighting during the light phase of the photoperiod, and larvae raised in the anesthetic tricaine. Interestingly, they show that none of these perturbations affected the OMR response in larvae. The absence of effect in tricaine-exposed fish, in particular, is most surprising given the teaching dogma that activity-dependent mechanisms are required to establish and/or fine tune brain circuits. The authors data calls this into question, making the relevance of the study very high. Overall, the study is well done, the figures are beautiful, and the experimental design is sound.

Specific comments:

1. The authors note that tricaine is a Na⁺ channel blocker, but they assess neuronal activity by examining Ca²⁺ levels. It would be worth explaining this connection in the text.
2. There are calcium-dependent action potentials that occur, and some have been identified in zebrafish. Though these tend to be in the minority (compared to sodium-based action potentials) the authors should speak to these. Could these channels be a mechanism that would allow circuits to form/develop in the presence of tricaine?
3. Please clarify the time course of recovery from tricaine exposure. The authors indicate that behavior recovered with a time course of 25 minutes, but neuronal recovery took an hour.

4. top of page 9 states 'we performed behavioral testing immediately after anesthetic washout' What is meant by 'washout'? Does it refer to removing the larvae from the tricaine solution? Or does it refer to biological washout? If it is the latter, how was that determined?

5. Figure 3b – what do the authors think is happening at the 24hr time point during stimulus presentation? The larvae show increased responsiveness with increase postexposure duration through 6-7hr, but then there is a decrease (at 24hr).

6. Please explain why the delay in response after lifetime tricaine was longer than the delay observed with a short tricaine exposure. Does this relate to penetration of the tricaine? Or could there be some synaptic depression/loss due to prolonged tricaine exposure that requires recovery?

7. The authors conclude that neuronal circuit development must be under genetic control. While this likely plays a role, what about endocrine control? Could that be involved as well?

8. In the discussion where the authors are citing examples of activity dependent circuit development, retinal waves should be mentioned.

9. Top of page 17 (discussion) notes that 'our perturbation is limited to the block of classical action potentials....' which is given as a comparison of cited studies that previously blocked 'trans-synaptic' mechanisms. While true, it is also likely the case that blocking voltage-gated sodium channels would also block synaptic transmission (secondary, downstream effect) as neurotransmitter would not be released. The authors should consider this as part of their discussion.

10. The authors results are truly intriguing and suggest that, in addition to genetic/genomic mechanisms, there may also be 'back up' mechanisms (calcium dependent action potentials for example) that are present and affect circuit development. These additional mechanisms are only revealed once traditional sodium channels are blocked.

11. Methods – please provide the specifics of pump system and the exchange of liquids surround the larvae when they were in agarose. What was the flow/exchange rate? Were the larvae in a static fluid or perfused? Etc.

Minor:

1. Figure 2b notes 'awake' fish. 'not anesthetized' would be a better term

2. Figure 3e is not mentioned in the text

3. First sentence of the discussion: 'we have developed a reversible lifetime block...' The block wasn't really 'lifetime'. 'developmental block' would be more accurate.

4. Page 16, 2nd paragraph (begins with 'in a subset of experiments...') – it is unclear what experiments this paragraph is referring to. Is it building on the previous paragraph?

Reviewer #4 (Remarks to the Author):

This paper presents exciting new data relating to the early development of neural circuits, and suggests that innate mechanisms of circuit formation supersede activity/experience-dependent processes, at least in the early critical periods. The paper is timely, and uses the versatile and exciting model species, zebrafish. It is novel and certainly worthy of publication. There are a couple of points

that I was not clear on, and I think that the authors should address before this manuscript is accepted.

1. The long term effects of tricaine - there is evidence for the effects on neuronal activation during 1h, but there is no data on long-term suppression of activity. How can the authors be sure of suppression over this time period? I.e., that no compensatory mechanisms occur? For instance, do they titrate the concentration of tricaine over the period?

2. The other problem conceptually is that the tricaine testing does not take place immediately after washout (as the authors concede - this would not be possible) -- function returns after a few hrs. It is possible, therefore, that what the authors are observing is a period of very fast/high plasticity. This would mirror fast adaptations seen (for example) in children's rapid recovery from hemispherectomy. Did the authors consider rapid recovery during critical periods?

Finally, as a minor comment, more details needed for the fish - were they fed at 5dpf? If so, how was this achieved in the dark/strobe conditions?

Response to Reviewers:

The Reviewers offer multiple recommendations to improve the paper, which we have fully taken into account in the revised manuscript.

Specifically, we have added whole-cell patch clamp experiments to our approach. This much more sensitive technology allowed us to confirm tricaine's efficacy in larval zebrafish, and further allowed us to verify the extent of silencing that is induced by tricaine both when delivered transiently and when applied persistently. We believe that the manuscript has improved significantly through the addition of this technology.

In the following, we provide a point-by-point response to all comments and suggestions. We have highlighted our answers and responses - as well as all the related changes in the manuscript - in blue.

Reviewer #1

The paper by Barabási and colleagues asks to which degree the development of circuits and behaviors in the zebrafish brain depends on neuronal activity, be it sensory evoked or spontaneous. They start by showing that a simple, visually driven behavior, the optomotor response (OMR), is present in larvae that were completely deprived of vision throughout development. Importantly, even during the very first testing trials after dark rearing, fish largely showed correct responses, indicating that they were able to perform this behavior without any "training". Next, in order to probe for a potential role of spontaneously generated activity in establishing the behavior, they raised zebrafish under tricaine, which, as they show using calcium imaging, abolishes all neuronal activity. Despite this, fish showed correct OMR behavior after only a very short period of drug washout. Tricaine raised fish eventually matched control levels, however, only after an extended period of drug washout. Finally, using calcium imaging in key brain regions involved in the OMR, they show that very shortly after drug washout, neurons show tuning for stimulus direction and other, more complex features, like evidence accumulation. The authors conclude that the circuits underlying these computations are assembled without neuronal spiking activity.

These are exciting results, which make a strong case in the old debate on the role of activity in the formation of neuronal circuits. While the paper very certainly won't close the case, it provides compelling evidence that, at least for the system used, neuronal activity is far less important than many researchers would have thought.

I have a few questions and comments, mostly minor, which might help improving the manuscript.

1. I am missing statistical tests for some of the data reported.

Following the suggestion by the Referee, we have added statistical tests to Fig. 3 b-d and Fig. 4 k. Briefly, these tests show that we consistently do see significant statistical differences between tricaine-reared and normally-reared animals; however, the effect sizes (difference in medians) are very small, see for example Fig. 4 k. We would like to thank the Referee for this suggestion and we think that these analyses strengthen the data in our manuscript.

2. In the introduction, the authors are slightly loose on the development of ocular dominance. “Binocular rivalry” is not exactly what drives this process, but rather describes the psychophysical phenomenon of rapidly alternating perceptual dominance of one or the other eye. Also, do they mean “ocular dominance formation” or formation of ocular dominance columns? The citations given here (9-11) are not exactly the correct ones. References 12 and 13 did not report structural abnormalities.

The reviewer is correct. We did not use the correct terminology. We have updated now the description in the text to provide a more accurate representation. We also made sure that the correct references are cited to support each claim.

3. Authors should clarify for how long dark reared fish were exposed to light (if at all) before going into the first behavioral trials. I am asking, since many experiments have shown that even brief exposure to stimuli can have effects on the development of behavior and response properties.

Thank you for highlighting this oversight – we now clarify that fish are kept in the dark until the moment behavioral experimentation begins. This is true both for dark-reared fish, as well as tricaine-reared fish, which we also now state later in the text.

4. Figure 2e, and others: colors are referred to as yellow and pink, and then salmon and gold elsewhere.

Thank you for catching this – referring to Figure 2e-i and 4e-i all colors should be salmon and gold. Elsewhere, we do use yellow (Figure 1) and pink (2b, 3d, 4l-m). We have now corrected this throughout.

5. Methods: “simulation” should read “stimulation”.

Done.

6. Fig. 4k: sure about the numbers for one hour and lifetime tricaine? Seems they are not matching between Figure and main text.

Done.

7. They state “Careful inspection of all cross sections and brain regions, under active and silenced conditions, did not reveal any clear differences.” But that seems to be at odds with other statements (e.g. Fig. 4b legend).

We have clarified this apparent discrepancy. While tricaine treatment leads to slightly extended ventricles and a few exceptionally bright neurons that could indicate ongoing apoptosis, (both features that capture the eye at first glance -Figure 4 b, c), a quantitative analysis of cell densities showed that there is no big difference between active and silenced brains. The remarkable similarity is much more apparent in the side by side volumetric comparison shown in Supplementary Figure 10 and Movie 3.

We make this clearer now in the text, and have also updated the legend of Figure 4b.

8. In several instances, mostly in the supplemental Figures, axis labels are missing or unclear, e.g. S Fig. 16.

We have added additional detail for supplemental figure 2, 11 and 16, where we labels were previously unclear or missing.

Reviewer #2

This manuscript uses a sodium-channel blocker to abolish the visible neuronal activity using calcium imaging during development of larval zebrafish. The goal is to differentiate between hardwired circuits (nature), and the need for sensory stimuli (nurture) to develop a proper OMR response. The manuscript is well-written and clear, the results support most of the claims. However, the extensive existing literature on the topic makes the results unsurprising, and this work merely incremental.

There have been many studies looking at the development of visual-driven behaviors in zebrafish (and many animal models), which are not discussed in enough detail in the introduction. Enucleation, dark rearing, lesion, or other approaches have been used for decades now, and it is unclear what this manuscript offers beyond the use of anesthetics for part of the development. The concept of “hardwired” genetically encoded modules has a long history, which the authors only mention at the end of their results and in the discussion, but should be present in the introduction. A lot of the existing literature is only mentioned in the discussion, and the studies of the Sumbre and Goodhill labs on early zebrafish visual development in the absence of stimuli are only mentioned in passing.

The use of tricaine is intriguing, but the use of calcium imaging could hide subthreshold activity, or sparse firing resulting in low signals lost in the noise, which could still be enough spontaneous activity for the circuits to develop on top of genetic information. It is also unclear what effect, if any, tricaine would have on the retinal circuits, which are essential for direction selectivity, and whose retinal waves have been reported to not depend on action potentials (see 10.1016/S0896-6273(00)81121-6). So, some of the tuning observed could still be activity dependent, but in the retinal circuits.

The reviewer is correct, calcium imaging will fail to detect most subthreshold activity dynamics. In fact, our hypotheses rely explicitly on the presence of all kinds of subthreshold activity, such as spontaneous release of vesicles, calcium dynamics unrelated to spikes and general metabolic activity. All of these are most likely critical for the formation, development and assembly of neural circuits - and many of these will not be picked up by calcium imaging, which is particularly effective in detecting spiking related activity patterns. We have now added to the discussion to make this more clear.

With respect to the formation and assembly of retinal circuitry in particular, we are largely agnostic and agree with the Referee that our tricaine-block presumably does not act on the retinal circuits, which do not rely on sodium spikes. The critical point, however, is that the RGCs do fire sodium spikes and these are blocked in our experiments. Therefore, even though the sensor (the retina) develops unblocked, it is decoupled from the brain and nonetheless orientation- and direction-selective tuning in retinofugal circuits seems to develop normally.

We would also like to point out that the retina itself is one of the brain regions that is least susceptible to activity-dependent changes and is generally assumed to be assembled on the basis of genetic information and transcriptional regulation, rather than activity-dependent mechanisms.

Electrophysiology would be necessary to assess how thoroughly tricaine blocks neuronal activity at the concentration used.

We would like to thank the Referee for this suggestion and we have now performed a series of patch-clamp recordings during tricaine wash-in and wash-out. These data and results have been incorporated into a new Figure (suppl. Fig. 17) which is now added to the manuscript. By measuring the frequency of evoked and spontaneous spikes, we obtained an estimate of how well tricaine blocks sodium spikes in the recorded cells, and by measuring the frequency of spontaneous EPSPs, we obtained a readout of the network activity surrounding the recorded neurons. We believe that these results are in excellent agreement with our imaging experiments: we find that it takes up to one hour of tricaine treatment for evoked and spontaneous spikes to disappear. Critically, also the frequency of evoked EPSPs (and their amplitudes, not shown) decrease within that time and remain at low baseline levels, which likely reflect spontaneous vesicle-fusion events. This is interesting and allows us to highlight the earlier point of the Referee that subthreshold activity (spontaneous release events, metabolic activity, etc.) are not blocked by tricaine.

Furthermore, the authors themselves acknowledge the side effects of the long-term anesthetic use, making any inference difficult with the data available.

We agree that long term exposure to tricaine results in side effects which can largely be explained by physiological or metabolic insults. We respectfully disagree that these side effects prohibit any inference or conclusion on the assembly of neural circuits. The strongest argument in favor of this assumption is that, while behavioral consequences are significant and pronounced, the effects on neural dynamics are negligible. We have added to the text to make this point clearer.

Reviewer #3

In this manuscript the authors describe their experiments, using zebrafish, that test the role of nerve cell activity in the development of a specific circuit. The authors chose to examine optomotor responses (OMRs) as this is a behavior that can be quantified and is due to specific cells in the pretectum, tectum, and hindbrain. The authors compare responses of zebrafish larvae raised in total darkness, larvae subjected to strobe lighting during the light phase of the photoperiod, and larvae raised in the anesthetic tricaine. Interestingly, they show that none of these perturbations affected the OMR response in larvae. The absence of effect in tricaine-exposed fish, in particular, is most surprising given the teaching dogma that activity-dependent mechanisms are required to establish and/or fine tune brain circuits. The authors data calls this into question, making the relevance of the study very high. Overall, the study is well done, the figures are beautiful, and the experimental design is sound.

Specific comments:

1. The authors note that tricaine is a Na⁺ channel blocker, but they assess neuronal activity by examining Ca²⁺ levels. It would be worth explaining this connection in the text.

We thank the reviewer for the suggestion, and we have added now clarification to the text.

2. There are calcium-dependent action potentials that occur, and some have been identified in zebrafish. Though these tend to be in the minority (compared to sodium-based action potentials) the authors should speak to these. Could these channels be a mechanism that would allow circuits to form/develop in the presence of tricaine?

We agree with the reviewer that calcium-dependent action potential occur in the fish and that they are likely not blocked by tricaine. However, we believe that most if not all recurrent network activity relies on the propagation of sodium spikes through classical axonal information pathways. It is extremely unlikely that calcium spikes by themselves could sustain such brain wide activity patterns. In fact, we see no evidence for any such activity under tricaine block. As such we believe that the existence of sodium spikes is a necessary condition for the emergence of calcium dependent action potentials, and that they would not occur in isolation. We now have added additional language to make this clearer in the discussion.

3. Please clarify the time course of recovery from tricaine exposure. The authors indicate that behavior recovered with a time course of 25 minutes, but neuronal recovery took an hour.

The reviewer is correct in pointing out this apparent discrepancy. To resolve this we point out that the neuronal recovery is not abrupt but occurs gradually, and with a time course of approximately 40-50 minutes. Importantly, individual neurons can emerge from anesthesia as

early as 5 minutes (individual gray points in Figure 4k). We suggest that the most basic behaviors, such as generic tail flicks and swim behavior, can be supported by this minimal set of sentinel neurons that become active at the earliest time points. This is of course hard to prove, but it shows that our interpretation is fully compatible with the neuronal and behavioral data.

4. top of page 9 states 'we performed behavioral testing immediately after anesthetic washout' What is meant by 'washout'? Does it refer to removing the larvae from the tricaine solution? Or does it refer to biological washout? If it is the latter, how was that determined?

We indeed utilize washout to refer to the moment in which fish are placed in anesthetic-free water, or when anesthetic-free water is pumped into the rig, as in the imaging studies. We now clarify this point in the highlighted sentence.

5. Figure 3b – what do the authors think is happening at the 24hr time point during stimulus presentation? The larvae show increased responsiveness with increase post exposure duration through 6-7hr, but then there is a decrease (at 24hr).

The drop in median bout frequency between 6-7 hours and 24 hours is not meaningful and just reflects the variance in the noisy behavioral data. Our analysis suggests that the bout frequency saturates at the 6/7 hour level, an interpretation that is compatible with the plotted data. The difference between the distributions of 6-7h and 24-25h is not statistically significant (Fig. 3b). Please note that we did find many significant statistical differences between tricaine-reared and normally-reared animals (e.g. Fig. 3 c, d). Thus, we believe that the statistical power of our analyses are sufficiently large overall.

6. Please explain why the delay in response after lifetime tricaine was longer than the delay observed with a short tricaine exposure. Does this relate to penetration of the tricaine? Or could there be some synaptic depression/loss due to prolonged tricaine exposure that requires recovery?

We believe that the prolonged recovery time after lifetime tricaine is due to extended clearing times of tricaine and not to an activity dependent, synaptic plasticity process. It is difficult to unequivocally prove this, but the remarkable match of the emergence of general neural activity with the emergence of directional tuning and integration time constants (Figure 4k,l,m) indicates all activity patterns come online with the same dynamics. This makes it unlikely that basic activity emerges first, which then shapes subsequent processes. Rather, this points to a universal block that gets gradually removed.

7. The authors conclude that neuronal circuit development must be under genetic control. While this likely plays a role, what about endocrine control? Could that be involved as well?

We apologize for this lack of clarity, we utilized “genetic control” as a shorthand for spiking independent developmental processes. We believe that there is a plethora of mechanisms which are all at play and contribute to the formation of neural circuit, including endocrine control, as suggested. We now note that cellular signalling should be considered in this process.

8. In the discussion where the authors are citing examples of activity dependent circuit development, retinal waves should be mentioned.

Thank you, we have now added language to the discussion that discusses calcium waves that can propagate in the absence of sodium spikes.

9. Top of page 17 (discussion) notes that ‘our perturbation is limited to the block of classical action potentials....’ which is given as a comparison of cited studies that previously blocked ‘trans-synaptic’ mechanisms. While true, it is also likely the case that blocking voltage-gated sodium channels would also block synaptic transmission (secondary, downstream effect) as neurotransmitter would not be released. The authors should consider this as part of their discussion.

Thank you — we have now added language to the discussion that addresses this point.

10. The authors results are truly intriguing and suggest that, in addition to genetic/genomic mechanisms, there may also be ‘back up’ mechanisms (calcium dependent action potentials for example) that are present and affect circuit development. These additional mechanisms are only revealed once traditional sodium channels are blocked.

We thank the Reviewer for this kind remark, and we have further highlighted this point in response to comment #2.

11. Methods – please provide the specifics of pump system and the exchange of liquids surround the larvae when they were in agarose. What was the flow/exchange rate? Were the larvae in a static fluid or perfused? Etc.

We have added a paragraph to the Methods section that describes the specifics of the fluid exchange during our functional imaging experiments.

Minor:

1. Figure 2b notes ‘awake’ fish. ‘not anesthetized’ would be a better term
2. Figure 3e is not mentioned in the text
3. First sentence of the discussion: ‘we have developed a reversible lifetime block...’ The block wasn’t really ‘lifetime’. ‘developmental block’ would be more accurate.

4. Page 16, 2nd paragraph (begins with 'in a subset of experiments...') – it is unclear what experiments this paragraph is referring to. Is it building on the previous paragraph?

We appreciate the attention to details. We have fixed these errors and provided additional context where necessary.

Reviewer #4

This paper presents exciting new data relating to the early development of neural circuits, and suggests that innate mechanisms of circuit formation supersede activity/experience-dependent processes, at least in the early critical periods. The paper is timely, and uses the versatile and exciting model species, zebrafish. It is novel and certainly worthy of publication. There are a couple of points that I was not clear on, and I think that the authors should address before this manuscript is accepted.

1. The long term effects of tricaine - there is evidence for the effects on neuronal activation during 1h, but there is no data on long-term suppression of activity. How can the authors be sure of suppression over this time period? I.e., that no compensatory mechanisms occur? For instance, do they titrate the concentration of tricaine over the period?

We show evidence for complete suppression of neuronal activity not only during the one hour after tricaine delivery; we also show that there is no activity after four days of constant tricaine exposure, when we image during the time period before washout. The fact that we find the strongest suppression of activity after a four-day constant exposure argues strongly for a consistent, complete and continuous action of tricaine.

Also, we replenish tricaine in the lifetime fish every twelve hours. This procedure was motivated by the finding that tricaine has limited stability in solution and degrades with a time constant of 18-24 hours. This was supported by behavioral evidence where we confirmed the half-life of the anesthetic's efficacy as shown in Supplemental Figure 16.

We now clarify these points in the Methods and point readers to the results in Supplemental Figure 16 when the treatment is first discussed.

2. The other problem conceptually is that the tricaine testing does not take place immediately after washout (as the authors concede - this would not be possible) -- function returns after a few hrs. It is possible, therefore, that what the authors are observing is a period of very fast/high plasticity. This would mirror fast adaptations seen (for example) in children's rapid recovery from hemispherectomy. Did the authors consider rapid recovery during critical periods?

The possibility that activity might emerge first in an unspecific fashion right after washout, and that the fish then experiences a brief period of rapid training and plasticity is indeed an interesting alternative to our main hypothesis. We now distinguish between those two alternatives more clearly in the results: we state that tuning of cells emerges at the same time as the appearance of first activity patterns, i.e. the cells emerge fully tuned. This finding allows us to reject the possibility that significant retraining takes place which would allow the cells to incrementally acquire their response properties through activity dependent mechanisms.

Finally, as a minor comment, more details needed for the fish - were they fed at 5dpf? If so, how was this achieved in the dark/strobe conditions?

The fish were not fed under any conditions, as this would introduce an additional source of variation between standard rearing, where feeding is easier, dark/strobe, where feeding becomes more difficult, and anesthetized fish, where feeding is impossible. This presents a trade-off between brain maturation and exhaustion of the yolk sac's nutrition, hence we chose to image and do behavior experiments at 6dpf. We now clarify these points, and other related to our treatments, in the Methods.

REVIEWERS' COMMENTS

Reviewer #1 (Remarks to the Author):

The authors have addressed all my comments.

Reviewer #2 (Remarks to the Author):

While I appreciate the efforts that the authors have made in addressing the other reviewers' questions, this does not qualitatively affect my appraisal of the work. We still think that the authors sidestep much of the existing literature, including to their rebuttal to this specific point in my previous review.

The electrophysiology results are welcome; however, they were done on a different cell type, with a rationale that is not convincing as the other experiments are focused on visual tuning, and the cerebellum is not looked at in the manuscript. Although, I understand it may be because of technical limitations which make patching Purkinje cells easier in zebrafish. Furthermore, the authors use twice the concentration of tricaine, as well as the addition of a paralytic (I assume α -bungarotoxin as it is not specified, but has been used in zebrafish previously), which may have effects on the neuronal dynamics and nAChR are present in the cerebellum of mammals and has been shown in the hindbrain of zebrafish. But the author confirm that subthreshold mechanisms are still present and could explain the results beyond genetic determinations of the circuits, which limits the scope and interest of this manuscript.

Reviewer #3 (Remarks to the Author):

The authors have replied to all comments from this reviewer. This is an excellent paper.

Reviewer #4 (Remarks to the Author):

The authors have done an excellent job addressing mine, and others', comments. Congratulations on this excellent work.

Reviewer #2 (Remarks to the Author):

While I appreciate the efforts that the authors have made in addressing the other reviewers' questions, this does not qualitatively affect my appraisal of the work. We still think that the authors sidestep much of the existing literature, including to their rebuttal to this specific point in my previous review.

The electrophysiology results are welcome; however, they were done on a different cell type, with a rationale that is not convincing as the other experiments are focused on visual tuning, and the cerebellum is not looked at in the manuscript. Although, I understand it may be because of technical limitations which make patching Purkinje cells easier in zebrafish. Furthermore, the authors use twice the concentration of tricaine, as well as the addition of a paralytic (I assume α -bungarotoxin as it is not specified, but has been used in zebrafish previously), which may have effects on the neuronal dynamics and nAChR are present in the cerebellum of mammals and has been shown in the hindbrain of zebrafish. But the author confirm that subthreshold mechanisms are still present and could explain the results beyond genetic determinations of the circuits, which limits the scope and interest of this manuscript.

Following the suggestion of this Referee from the first round of revisions, we have now added references to the Introduction (1) that speak to the history of the idea of 'hard-wired' (genetically encoded) circuits (lines 43-44) and (2) to papers that report developmental perturbation experiments (49-51).

The Referee is correct in that Purkinje cells are comparatively easy to patch because they are accessible due to their superficial location. We used 1 mg/ml bungarotoxin, see line 526.

Please note that the higher concentration of tricaine (200 μ g/ml) was only used in the wash-in experiments. For the wash-out experiments, we used 100 μ g/ml - the same concentration as in our behavioral and imaging experiments for 6 dpf fish.

We chose to use 200 μ g/ml for the wash-in experiments because we noticed that, for technical reasons, fish were not being anesthetized at the same rate as in our imaging experiments following wash-in. This is because the patch-clamp recording chamber is much smaller than the dishes used in the 2-photon microscope or during behavioral experiments. Thus, the agarose surrounding the animal, which effectively acts as a slowly-releasing reservoir containing regular fish water, takes up proportionally more volume and the net tricaine concentration surrounding the animal is considerably diluted. At the same time, less liquid can be exchanged in the patch clamp setup, because extended perfusion causes vibrations and recordings can be lost.

For that reason, we conducted a set of behavioral experiments and found that when we washed in 200 μ g/ml tricaine in the patch setup, animals became unresponsive to touch after a few minutes, comparable to our behavioral experiments. This has been specified in lines 527-532.

Finally, we would like to emphasize again that our hypotheses rely explicitly on the presence of all kinds of subthreshold activity, including spontaneous release of vesicles, calcium dynamics unrelated to spikes and metabolic activity. We believe these mechanisms are likely important for the formation, development and assembly of neural circuits. We have emphasized this point even more now in the Discussion (lines 389f).